# Effect of stroke early supported discharge on length of hospital stay: analysis from a national stroke registry

Rebecca J Fisher,[1] Adrian Byrne  ,[1] Niki Chouliara,[1] Sarah Lewis,[2] Lizz Paley,[3] Alex Hoffman,[3] Anthony Rudd,[3] Thompson Robinson,[4] Peter Langhorne,[5] Marion Walker[1]

[1]Division of Rehabilitation, Ageing and Wellbeing, University of Nottingham, Nottingham, UK
[2]Division of Epidemiology and Public Health, University of Nottingham, Nottingham, UK
[3]Department of Population Health Sciences, King's College London, London, UK
[4]Department of Cardiovascular Sciences, University of Leicester, Leicester, UK
[5]Institute of Cardiovascular & Medical Sciences, University of Glasgow, Glasgow, UK

**Correspondence to**
Dr Rebecca J Fisher;
rebecca.fisher@nottingham.ac.uk

## ABSTRACT

**Objective** The first observational study to investigate the impact of early supported discharge (ESD) on length of hospital stay in real-world conditions.

**Design** Using historical prospective Sentinel Stroke National Audit Programme (SSNAP) data (1 January 2013–31 December 2016) and multilevel modelling, cross-sectional (2015–2016; 30 791 patients nested within 55 hospitals) and repeated cross-sectional (2013–2014 vs 2015–2016; 49 266 patients nested within 41 hospitals) analyses were undertaken.

**Setting** Hospitals were sampled across a large geographical area of England covering the West and East Midlands, the East of England and the North of England.

**Participants** Stroke patients whose data were entered into the SSNAP database by hospital teams.

**Interventions** Receiving ESD along the patient care pathway.

**Primary and secondary outcome measures** Length of hospital stay.

**Results** When adjusted for important case-mix variables, patients who received ESD on their stroke care pathway spent longer in hospital, compared with those who did not receive ESD. The percentage increase was 15.8% (95% CI 12.3% to 19.4%) for the 2015–2016 cross-sectional analysis and 18.8% (95% CI 13.9% to 24.0%) for the 2013–2014 versus 2015–2016 repeated cross-sectional analysis. On average, the increased length of hospital stay was approximately 1 day.

**Conclusions** This study has shown that by comparing ESD and non-ESD patient groups matched for important patient characteristics, receiving ESD resulted in a 1-day increase in length of hospital stay. The large reduction in length of hospital stay overall, since original trials were conducted, may explain why a reduction was not observed. The longer term benefits of accessing ESD need to be investigated further.

**Trial registration number** http://www.isrctn.com/ISRCTN15568163.

## Strengths and limitations of this study

- ► First observational study comparing length of hospital stay in real-world settings for patients accessing early supported discharge (ESD) compared with those that did not.
- ► Use of datasets from the Sentinel Stroke National Audit Programme (SSNAP), permitting sample sizes of 30 791 and 49 266 patients.
- ► Accounts for the variation that exists between patients clustered within their admitting hospitals by employing a multilevel modelling approach and reinforces group comparisons using a propensity score matching process.
- ► Unable to account for residual confounding.
- ► Did not investigate the longer term benefits of accessing ESD.

## INTRODUCTION

Stroke is one of the main causes of adult disability, and there is strong research evidence that provision of stroke specialist rehabilitation enhances recovery.[1] Stroke early supported discharge (ESD) is a multidisciplinary team intervention aimed at facilitating discharge from hospital and providing stroke specialist rehabilitation at home.[2] Based on cumulative evidence from clinical trials, stroke care guidelines in the UK and worldwide recommend the provision of ESD as part of an evidence-based stroke care pathway.[3–8]

In many high-income countries, ESD has not been well developed in practice with a lack of large-scale implementation.[9] In contrast, in the UK, considerable efforts have been made to implement ESD, although types of service differ across the country, and in some regions, ESD is still not offered at all.[10] Where ESD has been delivered, it is unclear whether benefits of the ESD intervention, as suggested in clinical trials, are achieved in practice.

One of the attractive features of the ESD intervention as reported in clinical trials has been reduction in length of hospital stay, with the most recent Cochrane systematic review suggesting a reduction of 6 days.[2] This reduction has contributed to the reported cost-effectiveness of combined stroke unit

care and ESD.[11] Small-scale evaluation of ESD operating in localised areas have suggested smaller, but significant, reductions in length of acute hospital stay, but it remains unknown what impact ESD is having over a larger scale.[12] Average length of hospital stay for all stroke patients discharged alive fell from 40 days in 2001 to 20 days in 2013 and has remained around this level since then.[13] This trend over time has meant more stroke unit beds are available for new patients, but it has also increased demand for patient care in the community posthospital discharge.

The Sentinel Stroke National Audit programme (SSNAP) is the national stroke register of England, Wales and Northern Ireland in which all acute admitting hospitals and postacute stroke teams are mandated to participate.[14] SSNAP has played a key role in monitoring performance and improving provision of acute stroke care. Collection of SSNAP data across the stroke care pathway now offers a unique opportunity to investigate the large-scale impact of postacute interventions such as ESD. The study reported here, and in a parallel paper exploring the effectiveness of ESD service models, investigates if trial-based benefits of ESD are realised in practice, with the aim to inform evidence based improvements.[15] Here, we focus on the impact of ESD on length of hospital stay.

## METHODS
### Study design
We present results from an observational cohort study (figure 1), conducted as part of an overall mixed method study.[16] Data access requests should be directed to Healthcare Quality Improvement Partnership as the data controller and SSNAP as the data provider. The study protocol including statistical analysis plan is available online.[16] We determined a priori a sample size of 21 760 patients for a study power of 80% to detect standardised effect sizes of 0.25 for each outcome.

In this paper, we have used two different study designs: (1) a cross-sectional analysis (SSNAP data from 2015 to 2016 time period) comparing length of hospital stay between patients who did and did not have ESD on their care pathway, with adjustment for hospital and individual confounding factors; and (2) repeated cross-sectional analysis using two sets of SSNAP data (2013–2014 and 2015–2016) to establish whether length of hospital stay changed over time and whether any change was attributable to ESD.

### Setting
Hospitals (and associated ESD services) were sampled across a large geographical area of England. The sampling strategy was devised in accordance with the overall mixed method study design and included all ESD services in specific regions of England.[16] Here we report findings from the quantitative investigation of ESD effect on length of hospital stay across West and East Midlands

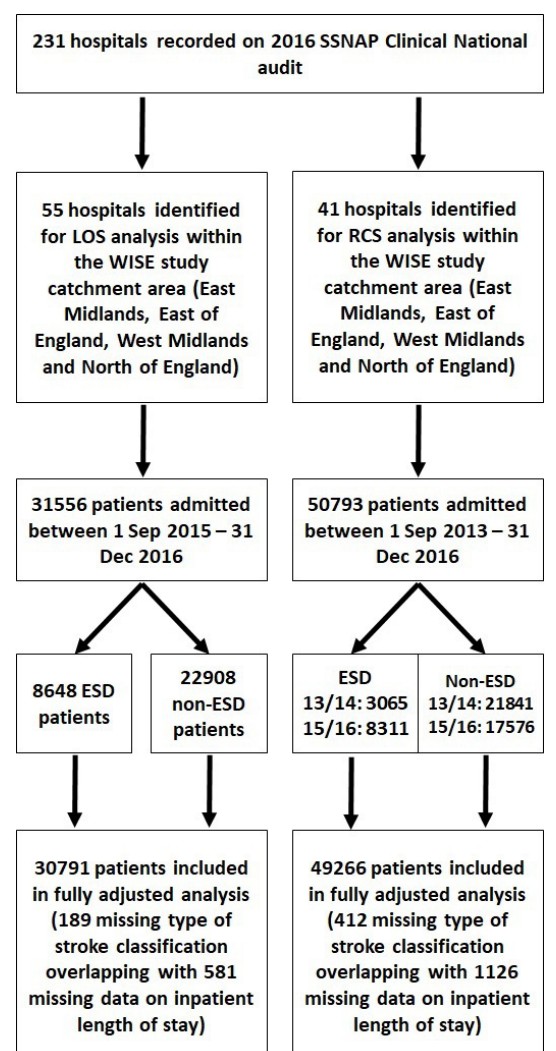

**Figure 1** Study design flow chart. ESD, early supported discharge; LOS, length of stay; RCS, repeated cross-sectional; SSNAP, Sentinel Stroke National Audit Programme; WISE, What is the Impact of Stroke Early supported discharge?

and East of England (across which a specific initiative to promote ESD was initiated in 2010) and the North of England, a region with a defined lack of ESD.[10 17] We have included all main referring hospitals within the defined geographical area as well as referring hospitals whose patients did (or did not) go onto receive ESD along their care pathway within the catchment area.

### Data sources and participants
Patient-level SSNAP data are entered by clinical teams onto a secure webtool with real-time data validations to ensure data quality.[14] Historical prospective clinical (patient-level) SSNAP data were obtained from the SSNAP team with permission from Healthcare Quality Improvement Partnership. The first study design involved 55 admitting hospitals and SSNAP data for all patients admitted during 1 September 2015–31 December 2016 (31 556 stroke patients). The second study design included 41 admitting hospitals, which were those of the 55 hospitals for which

we could obtain SSNAP data for all patients admitted during two time periods, 1 September 2013–31 December 2014 and 1 September 2015–31 December 2016 (50 793 stroke patients).

## Outcomes

Length of hospital stay was defined as the total length of hospital stay per patient (as an inpatient) from arrival at the admitting hospital (or time of stroke onset, if already an inpatient) to the time of discharge from the last inpatient setting, including time spent at any other inpatient setting. Owing to the positive skewness of the underlying distribution, length of hospital stay was natural log transformed before undertaking the statistical modelling, and all effect sizes were exponentiated afterwards to aid interpretation.

## Confounding factors

In order to investigate the effect of ESD on length of hospital stay, we identified a need to control for the overall standard of hospital care and also the influence of provision of social care. At the hospital level, we included two confounding variables: a hospital SSNAP rating score and a measure of delayed transfers of care from hospital, derived from the Adult Social Care Outcomes Framework (ASCOF).

The hospital rating scores used in this study were an overall quality rating for each hospital obtained from SSNAP (total key indicator score derived across 10 domains of stroke care with adjustments made for case ascertainment levels and the quality of data submitted to SSNAP).[18] SSNAP produced performance rating scores for admitting hospitals during the periods of interest, and we used the score produced during the latter part of the period of interest for each analysis.[19] This score for each admitting hospital was used as an indication of the overall standard of inpatient care after the onset of stroke. The ASCOF data report the average daily rate of delayed transfers of care per 100 000 population aged 18 years and over at local authority level. Data were aggregated up to National Health Service Trust level, using averages where multiple local authorities were associated with one National Health Service Trust. Again we focused on the scores produced during the latter part of the period of interest in line with the SSNAP hospital performance ratings.[20]

In order to account for comparison between different groups of individual patients, we also included variables at the patient level. These were stroke patient characteristics, reflecting validated stroke case-mix models and collected as part of the SSNAP data set. These were age at admission, sex, prestroke independence, cardiovascular comorbidities, National Institutes of Health Stroke Scale (NIHSS) score on admission, type of stroke and modified Rankin Scale score at discharge from hospital.[21 22]

## Statistical analyses

We employed a linear multilevel model framework for both study designs whereby patients (level 1) were clustered within admitting hospitals (level 2) in an approach consistent with previous observational studies of this type.[15 21–24] Within our multilevel models, we adjusted for a range of patient and hospital characteristics as covariates. We chose multilevel modelling to evaluate the effect of ESD on length of hospital stay as it could accommodate and appreciate the variation that existed within and between different admitting hospitals. Furthermore, the intraclass correlation coefficient was calculated as a measure of proportion of the total variance in length of hospital stay, which was attributable to variance within admitting hospitals as opposed to between admitting hospitals.

The adequacy of different statistical models was compared using the log-likelihood, Akaike Information Criterion and Bayesian Information Criterion values from single level and multilevel regression models with multilevel preferable on each occasion. Multicollinearity was investigated by examining variance inflation factor scores of all predictor variable sets and was found not to be an issue. Covariate linearity was examined by checking the consistency of a linear trend in relation to each outcome variable. Regarding the impact of missing data, we assessed the mean change in the outcome variable between the ESD and non-ESD groups in relation to missing predictor data (which amounted to a loss of 0.6% of the total sample size); no substantial differences were found as we observed a mean difference of less than 0.3%.

## Analysis of cross-sectional data from 2015 to 2016

A quasiexperimental cross-sectional design was used in which patients who received ESD on their care pathway were compared with patients that had not (non-ESD). Hence, across 55 admitting hospitals covering the 1 September 2015–31 December 2016 data period, we investigated the effect of ESD being on the care pathway (yes/no) for patients who had a stroke on their total length of hospital stay from admitting hospital until being discharged from the last inpatient setting.

## Analysis of repeated cross-sectional data from 2013 to 2014 and from 2015 to 2016

This analysis was to establish whether length of hospital stay changed over time, that is, between the earlier (1 September 2013–31 December 2014) and later time periods (1 September 2015–31 December 2016), and whether any change was attributable to ESD, as ESD provision was higher in the later time period.[10] Patients formed distinct groups over the two time periods as they were either admitted in the 2013–2014 time period or the 2015–2016 time period with no patient covering both time periods.[25] The same multilevel model approach was used as in the cross-sectional analysis and also incorporating an effect of time (13/14 or 15/16) and an interaction

between time and the effect of ESD, which modelled whether the effect of ESD was similar in the two time periods. The main effect of receiving ESD (with adjustment for time) indicated the effect of ESD on length of hospital stay in the first time period. An interaction between time and ESD would indicate whether there was any difference in the effect of ESD on length of hospital stay between the first and second time periods.

### Sensitivity analysis using propensity score matching

Since it was important to ensure that the effects of ESD were not the result of potential confounding factors, we also used an alternative method of controlling for confounders, that is, propensity score matching, as a sensitivity analysis. Propensity score matching involved matching patients with ESD on their pathway with similar patients who did not, based on their patient characteristics. In this case, we explored two alternative approaches to propensity score matching, first matching patients with ESD to patients without ESD in the same hospital, that is, within-admitting hospital matching.[26] This resulted in a truncated sample size as only hospitals with both ESD and non-ESD patients could contribute to this first matching analysis.

Second, we then allowed the matching to occur with similar patients without ESD in different hospitals, that is, between-admitting hospital matching.[27] This second approach enabled us to use a larger sample size as all hospitals could contribute to the probability estimate of whether a patient received ESD or not regardless of what their patients actually received. In order to ensure greater accuracy in the latter analysis, we used a stratified matching process so that patients were only matched to patients in a hospital with a similar proportion of patients having access to ESD.

A two-tailed significance level of 0.05 was used in all hypothesis tests. We carried out the majority of analyses using Stata/SE V.15.1, and the R package Matching was used to undertake the between-admitting hospital propensity score matching.

### Patient and public involvement

The design and conduct of this study was informed through discussion with the Nottingham Stroke Partnership group. Two stroke survivor members were part of the study steering group and advised on a lay summary of our study findings.

## RESULTS
### Analysis of cross-sectional data from 2015 to 2016

Table 1 presents descriptive statistics of the patient and admitting hospital level variables and the median number of days spent as an inpatient disaggregated by patients who received ESD (ESD; n=8648) or not (non-ESD; n=22 908) on the care pathway. Compared with patients who did not receive ESD, those who did were younger and more likely to be independent (lower premorbid modified Rankin

| Patient characteristics | ESD (n=8648) | Non-ESD (n=22 908) |
|---|---|---|
| **Age, years** | | |
| <60 | 1422 (16.4) | 3674 (16.0) |
| 60–69 | 1692 (19.6) | 3932 (17.2) |
| 70–79 | 2646 (30.6) | 6080 (26.5) |
| 80–89 | 2373 (27.4) | 6873 (30.0) |
| >89 | 515 (6.0) | 2349 (10.3) |
| **Gender** | | |
| Male | 4869 (56.3) | 11 785 (51.4) |
| **Already inpatient at time of stroke** | | |
| Yes | 264 (3.1) | 995 (4.3) |
| **Congestive heart failure prior to admission** | | |
| Yes | 348 (4.0) | 994 (4.3) |
| **Hypertension prior to admission** | | |
| Yes | 4743 (54.9) | 12 299 (53.7) |
| **Atrial fibrillation prior to admission** | | |
| Yes | 1370 (15.8) | 4016 (17.5) |
| **Diabetes prior to admission** | | |
| Yes | 1770 (20.5) | 4834 (21.1) |
| **Stroke/TIA prior to admission** | | |
| Yes | 2075 (24.0) | 6100 (26.6) |
| **Modified Rankin Scale score before stroke** | | |
| 0 | 5688 (65.8) | 12 796 (55.9) |
| >0 | 2960 (34.2) | 10 112 (44.1) |
| **NIHSS score on arrival** | | |
| 0 | 1041 (12.0) | 3267 (14.3) |
| 1–5 | 4640 (53.7) | 11 252 (49.1) |
| 6–14 | 2253 (26.1) | 5596 (24.4) |
| 15–24 | 636 (7.4) | 2322 (10.1) |
| >24 | 78 (0.9) | 471 (2.1) |
| **Type of stroke** | | |
| Infarction | 7804 (90.8) | 20 772 (91.2) |
| Primary intracerebral haemorrhage | 789 (9.2) | 2002 (8.8) |
| **Modified Rankin Scale score at inpatient discharge** | | |
| 0 | 1046 (12.1) | 4016 (17.5) |
| 1 | 2259 (26.1) | 5024 (21.9) |
| 2 | 1928 (22.3) | 3420 (14.9) |

Continued

**Table 1** Continued

| Patient characteristics | ESD (n=8648) | Non-ESD (n=22 908) |
|---|---|---|
| 3 | 2189 (25.3) | 3838 (16.8) |
| 4 | 1118 (12.9) | 4154 (18.1) |
| 5 | 108 (1.3) | 2456 (10.7) |
| Median (IQR) total length of hospital stay in days | 6.9 (2.9–18.8) | 6.0 (2.2–24.8) |

Values are numbers (percentages) unless stated otherwise.
ESD, early supported discharge; NIHSS, National Institutes of Health Stroke Scale; TIA, transient ischemic attack.

Scale score) before stroke as well as more independent at inpatient discharge (lower postmorbid modified Rankin Scale score). ESD patients were also more likely to have been admitted to a hospital with a higher SSNAP score rating and higher rate of delayed transfers.

Before adjustment, ESD patients had a longer length of hospital stay than non-ESD patients (median 6.9 (IQR: 2.9–18.8) compared with 6.0 days (IQR: 2.2–24.8)). Following multilevel model analysis, after controlling for patient and hospital level characteristics, the mean length of stay for patients who received ESD along their care pathway compared with those who did not, remained significantly longer by 15.8% (95% CI 12.3% to 19.4%) (table 2). Similarly, from the sensitivity analysis using propensity score matching, patients who received ESD on their care pathway, on average, stayed in hospital longer (when controlling for all other patient characteristics, and hospital SSNAP and ASCOF scores) (table 3). The within-admitting hospital propensity score matching process (n=10 449) suggested ESD patients stayed in hospital longer by an average of 8.7% (95% CI 4.8% to 12.6%). The between-admitting hospital propensity score

matching process (n=30 791) found that ESD patients stayed in hospital longer by an average of 7.3% (95% CI 3.0% to 11.8%).

### Analysis of repeated cross-sectional data from 2013 to 2014 and from 2015 to 2016

Table 4 presents descriptive statistics of the patient and admitting hospital level variables used in the repeated cross-sectional analysis, as well as the median number of days spent as an inpatient disaggregated by patients who received ESD (n=11 376) or not (non-ESD; n=39 417) on the care pathway across the two time periods. This descriptive repeated cross-sectional analysis showed that the numbers of patients who received ESD significantly increased over the two time periods. Moreover, across the two time periods, ESD patients were younger, more likely to be male and more likely to have premorbid independence, as evidenced by a lower modified Rankin Scale score. Before adjustment, ESD patients were recorded as having a longer length of stay in hospital than non-ESD patients (median 7.7 (IQR: 3.2–23.8) compared with 6.3 days (IQR: 2.5–22.1) in 2013–2014 and 6.8 (IQR: 2.9–18.5) compared with 5.7 days (IQR: 2.1–24.6) in 2015–2016).

Table 5 presents the results from the linear multilevel model. Again, we controlled for patient characteristics, SSNAP admitting hospital score and average ASCOF rates for each admitting hospital. There was no significant interaction between receiving ESD and time period (p=0.79) so that the effect of ESD was similar in both time periods, despite more patients accessing ESD over time. Therefore, the main effect of ESD across both time periods was estimated to increase length of hospital stay by an average of 18.8% (95% CI 13.9% to 24.0%) compared with patients who did not receive ESD on their care pathway. Furthermore, the main effect of the 2015–2016 time period compared with the 2013–2014 time period

**Table 2** Association between having received ESD on the care pathway and inpatient length of stay 2015–2016

| Inpatient length of stay models | Unadjusted | | Adjusted* | |
|---|---|---|---|---|
| Patients | 30 975 | | 30 791 | |
| Admitting hospitals | 55 | | 55 | |
| Patients per admitting hospital | | | | |
| Min | 27 | | 26 | |
| Mean | 563.2 | | 559.8 | |
| Max | 1331 | | 1310 | |
| Intraclass correlation coefficient | 0.05 | | 0.08 | |
| **ESD on the care pathway** | **Coefficient (95% CI)** | **P value** | **Coefficient (95% CI)** | **P value** |
| Received ESD | 0.08 (0.04 to 0.12) | <0.001 | 0.15 (0.12 to 0.18) | <0.001 |

Model coefficients are on the natural log scale; significant results were back-transformed, that is, exponentiated to obtain percentage change in length of hospital stay (reported in text).
*Adjusted for age, sex, prestroke independence, comorbidities, NIHSS score on admission, type of stroke and modified Rankin Scale score at discharge from hospital (patient level); ASCOF rate and hospital SSNAP rating score (admitting hospital level).
ASCOF, Adult Social Care Outcomes Framework; ESD, early supported discharge; NIHSS, National Institutes of Health Stroke Scale; SSNAP, Sentinel Stroke National Audit Programme.

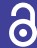

**Table 3** Matched analysis between patients who received ESD on the care pathway and similar patients who did not 2015–2016*

| Inpatient matching models | Between† | | Within† | |
|---|---|---|---|---|
| Patients | 30 791 (8593 ESD+22 198 non-ESD) | | 10 449 (4572 ESD+5877 non-ESD) | |
| Admitting hospitals | 55 | | 14 | |
| **ESD on the care pathway** | **Coefficient (95% CI)** | **P value** | **Coefficient (95% CI)** | **P value** |
| Received ESD | 0.07 (0.03 to 0.11) | <0.001 | 0.08 (0.05 to 0.12) | <0.001 |

Model coefficients are on the natural log scale; significant results were back-transformed, that is, exponentiated to obtain percentage difference in length of hospital stay (reported in text).
*This is a sensitivity analysis using propensity score matching to complement the multilevel model results presented in table 2. Between means patients in all hospitals were matched even if the hospitals did not offer ESD, whereas within means only hospitals with patients who did and did not receive ESD were matched.
†Adjusted for age, sex, prestroke independence, comorbidities, NIHSS score on admission, type of stroke and modified Rankin Scale score at discharge from hospital (patient level).
ESD, early supported discharge; NIHSS, National Institutes of Health Stroke Scale.

was found to significantly reduce the length of hospital stay by an average of 14.0% (95% CI 12.2% to 15.9%).

The degree of clustering was low in both adjusted multilevel models (adjusted intraclass correlation coefficients, were 0.08, 0.08 respectively) implying that the total variance we observed in the length of hospital stay data was more greatly accounted for by the patients than by the admitting hospitals.

## DISCUSSION

This study has investigated the effect of ESD on length of hospital stay in real-world settings, with the aim to inform evidence-based improvements in stroke care. Unadjusted findings indicated that patients who had a stroke who received ESD spent approximately 1 day longer in hospital than all other patients; however, those who received ESD differed in characteristics from those who did not. With adjustment using two different designs and alternative analyses for the differing case-mix, length of hospital stay remained longer for those who received ESD on their care pathway. This contrasts with findings from original ESD randomised controlled trials, in which a reduction in length of hospital stay was reported.[2]

This is an important finding, particularly when considering implementation of ESD in practice. The promise of reducing length of hospital stay made ESD attractive with regard to cost savings and is likely to have contributed to widespread implementation in England.[10 11] The economic implications of ESD in the real world have previously been investigated through simulation modelling (informed by SSNAP data similar to data used here).[28] The authors reported an estimated mean length of hospital stay for ESD patients that was 8 days less than non-ESD patients and thereby derived cost savings associated with modelling increased use of ESD.

However, an important difference with that economic analysis and the study reported here relates to the fact that we used actual and total inpatient length of stay data combining different inpatient settings where applicable

on a per-patient basis. Analysis showed that ESD patients were younger and more likely to have a lower modified Rankin Scale score at inpatient discharge. The simulation modelling findings drew on sample statistics derived from patient distributions for each separate inpatient setting stratified by age category and modified Rankin Scale score, and their results were driven by assuming more severe stroke survivors received ESD. What our study adds is that when comparing ESD and non-ESD patient groups matched for important patient characteristics, receiving ESD does not relate to a reduction in length of hospital stay.

Another important consideration is that average length of hospital stay for stroke survivors overall has reduced dramatically over the last 20 years, the latter decade of which our study captures between the two cohorts in this paper (ie, 2013–2014 and 2015–2016).[13] This makes it less likely for an intervention like ESD to have an effect (in comparison to the years 1997–2004 when the original UK clinical trials were conducted).[2] That ESD in this study was associated with an average increase in length of stay of 1 day could be interpreted as the need for additional time to facilitate the transfer of care to the ESD team, rather than stroke survivors simply leaving hospital without support (if ESD was not available). Previous studies have also highlighted transfer problems relating to lack of joint working between health and social care.[29–31] Our parallel paper added to this debate by highlighting the importance of access to a social worker as part of the ESD team.[15]

What was not possible to investigate in this study was whether access to ESD (despite an additional day in hospital) was associated with improved patient outcomes over the longer term. Recent observational studies in Sweden suggest patient and caregiver benefits related to provision of ESD in regular clinical practice, in line with our previous study findings from England.[32 33] We suggest routine collection of additional validated patient outcome measures (eg, measuring activities of daily

| Table 4 Patient characteristics from 2013 to 2014 and from 2015 to 2016 cohorts | | |
|---|---|---|
| Patient characteristics | ESD (n=11376) | Non-ESD (n=39417) |
| Age, years | | |
| <60 | 1847 (16.2) | 6212 (15.8) |
| 60–69 | 2239 (19.7) | 6647 (16.9) |
| 70–79 | 3467 (30.5) | 10543 (26.8) |
| 80–89 | 3126 (27.5) | 12087 (30.7) |
| >89 | 697 (6.1) | 3928 (10.0) |
| Gender | | |
| Male | 6352 (55.8) | 19959 (50.6) |
| Already inpatient at time of stroke | | |
| Yes | 361 (3.2) | 1766 (4.5) |
| Congestive heart failure prior to admission | | |
| Yes | 481 (4.2) | 1558 (4.0) |
| Hypertension prior to admission | | |
| Yes | 6261 (55.0) | 21313 (54.1) |
| Atrial fibrillation prior to admission | | |
| Yes | 1840 (16.2) | 7299 (18.5) |
| Diabetes prior to admission | | |
| Yes | 2345 (20.6) | 8074 (20.5) |
| Stroke/TIA prior to admission | | |
| Yes | 2740 (24.1) | 10482 (26.6) |
| Modified Rankin Scale score before stroke | | |
| 0 | 7389 (65.0) | 22847 (58.0) |
| >0 | 3987 (35.1) | 16570 (42.0) |
| NIHSS score on arrival | | |
| 0 | 1430 (12.6) | 6884 (17.5) |
| 1–5 | 6020 (52.9) | 18802 (47.7) |
| 6–14 | 2955 (26.0) | 9162 (23.2) |
| 15–24 | 861 (7.6) | 3833 (9.7) |
| >24 | 110 (1.0) | 736 (1.9) |
| Type of stroke | | |
| Infarction | 10315 (91.2) | 35989 (92.1) |
| Primary intracerebral haemorrhage | 998 (8.8) | 3079 (7.9) |
| Modified Rankin Scale score at inpatient discharge | | |
| 0 | 1440 (12.7) | 8474 (21.5) |
| | | Continued |

| Table 4 Continued | | |
|---|---|---|
| Patient characteristics | ESD (n=11376) | Non-ESD (n=39417) |
| 1 | 2928 (25.7) | 8649 (21.9) |
| 2 | 2562 (22.5) | 5456 (13.8) |
| 3 | 2817 (24.8) | 6090 (15.5) |
| 4 | 1471 (12.9) | 6733 (17.1) |
| 5 | 158 (1.4) | 4015 (10.2) |
| Time period | | |
| 2013–2014 | 3065 (26.9) | 21841 (55.4) |
| 2015–2016 | 8311 (73.1) | 17576 (44.6) |
| Median (IQR) total length of hospital stay in days | | |
| 2013–2014 | 7.7 (3.2–23.8) | 6.3 (2.5–22.1) |
| 2015–2016 | 6.8 (2.9–18.5) | 5.7 (2.1–24.6) |

Values are numbers (percentages) unless stated otherwise.
NIHSS, National Institutes of Health Stroke Scale; TIA, transient ischemic attack.

living, general health/mood and quality of life) at longer follow-up periods in national stroke audits or registries is required.[2 34 35]

Like other observational studies, this study had its limitations. Although numbers of patients were sufficient for the analysis undertaken, by focusing on specific regions of England, transferability of findings could be questioned, particularly outside England. Other limitations include potential sources of residual confounding and also reliance on SSNAP data being accurately reported by hospital teams. Length of hospital stay data are inherently variable, and while we tried to include a proxy measure for other sources of delays in discharge, analysis would have benefited from social care related patient level variables.[28 32] Patients also experienced a series of hospital stays (eg, transfer between acute and rehabilitation wards), relying on accurate transfer of SSNAP records between hospitals and resulting in us focusing on total length of hospital stay and ESD provision on a care pathway.[36] What this means is that the impact of ESD on a particular hospital may not be apparent, which may be of more interest to providers of hospital services.

## CONCLUSION
Original clinical trials of ESD were conducted across the world, and implementation of ESD is recommended in many countries' stroke guidelines.[4–9] This study investigated the impact of ESD in real-world settings and focused on length of hospital stay. ESD was not associated with a reduction in length of hospital stay as previously reported in clinical trials, although the increased average length of stay was just 1 day. This highlights the importance of investigating whether trial-based outcomes of interventions are realised in the real world. What

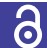

| Table 5 | Association between having received ESD on the care pathway and inpatient length of stay 2013–2016 | | | |
|---|---|---|---|---|
| **Inpatient length of stay models** | **Unadjusted** | | **Adjusted*** | |
| Patients | 49 667 | | 49 266 | |
| Admitting hospitals | 41 | | 41 | |
| Patients per admitting hospital | | | | |
| Min | 115 | | 106 | |
| Mean | 1211.4 | | 1201.6 | |
| Max | 2512 | | 2446 | |
| Intraclass correlation coefficient | 0.04 | | 0.08 | |
| **ESD on the care pathway** | **Coefficient (95% CI)** | **P value** | **Coefficient (95% CI)** | **P value** |
| Received ESD | 0.14 (0.09 to 0.20) | <0.001 | 0.17 (0.13 to 0.22) | <0.001 |
| 2015–2016 time period | −0.06 (-0.09 to -0.04) | <0.001 | −0.15 (-0.17 to -0.13) | <0.001 |
| Received ESD * 2015–2016 time period | −0.05 (−0.11 to 0.02) | 0.162 | 0.01 (−0.04 to 0.06) | 0.790 |

Model coefficients are on the natural log scale; significant results were back-transformed, that is, exponentiated to obtain percentage change in length of hospital stay (reported in text).

*Adjusted for age, sex, prestroke independence, comorbidities, NIHSS score on admission, type of stroke and modified Rankin Scale score at discharge from hospital (patient level); ASCOF rate and hospital SSNAP rating score (admitting hospital level).

ASCOF, Adult Social Care Outcomes Framework; ESD, early supported discharge; NIHSS, National Institutes of Health Stroke Scale; SSNAP, Sentinel Stroke National Audit Programme.

remains to be investigated and reported are the benefits of accessing ESD over the longer term. Also by focusing analysis on patients who would have been eligible, but did not receive ESD, we have highlighted an important gap in service provision.

**Correction notice** This article has been corrected since it first published. The provenance and peer review statement has been included.

**Acknowledgements** We would like to thank the many people and organisations participating in Sentinel Stroke National Audit Programme (SSNAP) and members of the SSNAP collaboration. We would also like to thank Trevor Gard and Frances Cameron, stroke survivors who were members of our study steering group.

**Contributors** All named authors made a substantial contribution across a number of areas including study design, data analysis, interpretation of findings, drafting/revising the manuscript and approving the final version for publication. All authors agree to be accountable for respective aspects of the work ensuring that questions related to the accuracy or integrity of any part of the work are appropriately investigated and resolved. RF (principal investigator) led in the original design of the study protocol, with input from MW, SL, TR, PL, AB and NC. LP, AH and AR are SSNAP collaborators and advised on access to audit data and appropriate use for research purposes. RF led running of the study, with AB and SL leading on statistical methodology, obtaining data from SSNAP, data analysis and interpretation, and article writing and approval; NC, MW, TR and PL advised RF on overall study design, data interpretation, article writing and approval; LP, AH and AR are SSNAP collaborators and were involved in data sharing agreements, advice on data handling, analysis, article writing and approval.

**Funding** This research was funded by the National Institute for Health Research (NIHR), Health Services and Delivery Research (HS&DR) Programme (16/01/17). RF is funded by the Stroke Association (TSA LECT 2016/01 Stroke Association HRH the Princess Margaret Senior Lecturer Award). TR is an NIHR senior investigator. The SSNAP is commissioned by the Healthcare Quality Improvement Partnership and funded by National Health Service England and the Welsh Government.

**Disclaimer** The views expressed are those of the authors and not necessarily those of the NHS, the NIHR or the Department of Health and Social Care.

**Competing interests** None declared.

**Patient and public involvement** Patients and/or the public were involved in the design, or conduct, or reporting, or dissemination plans of this research. Refer to the Methods section for further details.

**Patient consent for publication** Not required.

**Ethics approval** The study protocol was approved by the University of Nottingham Ethics Committee and the UK Healthcare Quality Improvement Partnership Data Access Request Group.

**Provenance and peer review** Not commissioned; externally peer reviewed.

**Data availability statement** Data may be obtained from a third party and are not publicly available. Data access requests should be directed to the Healthcare Quality Improvement Partnership as the data controller and the Sentinel Stroke National Audit Programme as the data provider.

**ORCID iD**
Adrian Byrne http://orcid.org/0000-0003-4887-2572

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
