## [Reviewer comments · BMJ Open]

ARTICLE DETAILS

TITLE (PROVISIONAL)	Effect of Stroke Early Supported Discharge on length of hospital stay: Analysis from a National Stroke Registry
AUTHORS	Fisher, Rebecca; Byrne, Adrian; Chouliara, Niki; Lewis, Sarah; Paley, Lizz; Hoffman, Alex; Rudd, Anthony; Robinson, Thompson; Langhorne, Peter; Walker, Marion

VERSION 1 – REVIEW

REVIEWER	Rohan Grimley Griffith University Australia
REVIEW RETURNED	07-Oct-2020

GENERAL COMMENTS	It was a pleasure to review this well performed and presented study investigating the real world impact of early supported discharge services on stroke hospital length of stay. The findings are relevant and of interest. Only minor revisions are suggested for improved clarity and accuracy. Although the statistical methods appear very sound and thorough, with appropriate sensitivity analysis, I do feel that expert statistical opinion should be sought regarding any issues with risks of back exponentiation to estimate effect sizes after natural log transformation of the data. Although the primary outcome of proportional change should be robust, whether the subsequent exponentiation of effect sizes “to aid interpretation” actually leads to a meaningful approximation of effect size is beyond my statistical knowledge. Would a technique such a median regression have better maintained integrity of the underlying data (even as a sensitivity analysis)? Some specific comments: Abstract Design you’ve omitted the verb: – “cross sectional ...and repeated cross sectional **what** were undertaken. Intro – salient, concise, appropriate Methods: I found the inclusion of referring hospitals outside of catchment areas confusing – did you mean – in addition to hospitals outside of these areas whose patients had accessed ESD within our geographical areas? The inclusion of sites outside of your defined
---

	geographical area currently seems somewhat arbitrary and requires some explanation. It is not clear why a different number of hospitals were included in the two studies – this is important to include. Results: The use of the word “increase” is intermittently used for both cross sectional comparison and also changes over time (eg Results line 31). Suggest use of descriptors such as “higher” or “greater” when referring to the difference in LOS between groups in the static cross sectional data such as reported line 36/39. Tables 1 and 4 could be simplified and carry the same information by only including those with the reported conditions (“yes rows”) I’m surprised that there doesn’t seem to be missing data in patient level characteristics – other than exclusions due to lack of inpatient LOS and stroke classification mentioned in Figure 1. It would be worth clarifying as this seems “too good to be true” for audit data. Discussion/conclusions: Overall well written and salient. Inclusion of an aim and subsequent conclusion regarding “to address inequality and inform evidence based improvements in stroke care” seems an over-call for this data and the analysis performed. There is little specific data, analysis or interpretation regarding inequality and I would suggest removing this from the aims and conclusions. On the other hand, the conclusion could be a little more blunt – this study provides concerning data that in contrast to one of the major aims and purported benefits of ESD, the length of stay in the real world appears to be longer! You seem to have under-emphasised this. This reinforces the importance of investigating whether the improved outcomes and client experience also differ between the trials and the real world, and perhaps re-investigating the cost implications.
--	---

REVIEWER	Sophie Lehnerer Charité University medicine Berlin, Germany
REVIEW RETURNED	13-Oct-2020

GENERAL COMMENTS	It would be nice to find the absolute number of inpatient days of the ESD and non-ESD group in the text and not only in the tables.
---

REVIEWER	Konstantakopoulou, Olympia National and Kapodistrian University of Athens, Nursing
REVIEW RETURNED	18-Oct-2020

GENERAL COMMENTS	This is a very well written paper aiming to investigate the impact of Early Supported Discharge (ESD) on length of hospital stay in real world conditions. I recommend it for publication without any revisions.
---

VERSION 1 – AUTHOR RESPONSE

Reviewer: 1

Comments to the Author

It was a pleasure to review this well performed and presented study investigating the real world impact of early supported discharge services on stroke hospital length of stay. The findings are relevant and of interest. Only minor revisions are suggested for improved clarity and accuracy.

Although the statistical methods appear very sound and thorough, with appropriate sensitivity analysis, I do feel that expert statistical opinion should be sought regarding any issues with risks of back exponentiation to estimate effect sizes after natural log transformation of the data. Although the primary outcome of proportional change should be robust, whether the subsequent exponentiation of effect sizes “to aid interpretation” actually leads to a meaningful approximation of effect size is beyond my statistical knowledge. Would a technique such as a median regression have better maintained integrity of the underlying data (even as a sensitivity analysis)?

We have raised Reviewer 1’s concerns here with one of the paper’s co-authors who is a professor of medical statistics. Their response tackles these concerns from three angles:

1. Is the model appropriate? The answer is (one could argue) that length of stay will have different distributions in different populations. We looked carefully at the distribution and a log transformation did transform to normality as examined by looking at the distribution in this case.

2. Is it appropriate to back transform by exponentiating? In general if the dependent variable is log transformed, the standard approach is to back transform (exponentiate if it’s natural log) the coefficient and interpret as the % increase.

3. Does this give a meaningful size of effect? It provides an effect which is more meaningful than the raw coefficient from the model (though this is also provided in Tables 2, 3 and 5) and enables you to compare between models and so on. However, we have appreciated that this may not be the most clinically useful measure of the effect of ESD, and hence have additionally provided estimates in terms of the median difference in Tables 1 and 4 as well as providing this detail in the text on page 10 (second paragraph in the *Results* section):

“Before adjustment, ESD patients had a longer length of hospital stay than non-ESD patients; median 6.9 (interquartile range: 2.9-18.8) compared to 6.0 days (interquartile range: 2.2-24.8)”

and on page 11 (third paragraph in the *Results* section):

“Before adjustment, ESD patients were recorded as having a longer length of stay in hospital than non-ESD patients; median 7.7 (interquartile range: 3.2-23.8) compared to 6.3 days (interquartile

range: 2.5-22.1) in 2013-2014, and 6.8 (interquartile range: 2.9-18.5) compared to 5.7 days (interquartile range: 2.1-24.6) in 2015-2016”

Some specific comments:

Abstract

Design you've omitted the verb: – “cross sectional ...and repeated cross sectional ****what**** were undertaken.

Thank you – we have inserted “analyses” within the *Design* section of the abstract as follows:

“cross-sectional (2015-2016; 30,791 patients nested within 55 hospitals) and repeated cross-sectional (2013-2014 vs 2015-2016; 49,266 patients nested within 41 hospitals) analyses were undertaken”

Intro – salient, concise, appropriate

Methods:

I found the inclusion of referring hospitals outside of catchment areas confusing – did you mean – in addition to hospitals outside of these areas whose patients had accessed ESD within our geographical areas? The inclusion of sites outside of your defined geographical area currently seems somewhat arbitrary and requires some explanation.

Yes we do acknowledge that some patients who ended up in our catchment area started their stroke journey outside our catchment area. Thanks to your comment here, we have simplified the last two lines of the *Setting* paragraph on page 5 into one line as follows:

“We have included all main referring hospitals within the defined geographical area as well as referring hospitals whose patients did (or did not) go onto receive ESD along their care pathway within the catchment area.”

It is not clear why a different number of hospitals were included in the two studies – this is important to include.

Thank you - we have attempted to make this clearer by amending the last line of the first paragraph in the *Data sources and participants* section on page 5 as follows:

“The second study design included 41 admitting hospitals, which were those of the 55 hospitals for which we could obtain SSNAP data for all patients admitted during two time periods, 1 Sep 2013 – 31 Dec 2014 and 1 Sep 2015 – 31 Dec 2016 (50,793 stroke patients).”

Results:

The use of the word “increase” is intermittently used for both cross sectional comparison and also changes over time (eg Results line 31). Suggest use of descriptors such as “higher” or “greater” when referring to the difference in LOS between groups in the static cross sectional data such as reported line 36/39.

Thank you – on foot of this comment we have made one change to the way we refer to the static 2015-2016 analysis as per your suggestion within second paragraph in the Results section on page 10:

“Following multilevel model analysis, after controlling for patient and hospital level characteristics, the mean length of stay for patients who received ESD along their care pathway compared to those who did not remained significantly longer by 15.8% (95% CI 12.3% - 19.4%) (Table 2)”

Tables 1 and 4 could be simplified and carry the same information by only including those with the reported conditions (“yes rows”)

Thank you – done as suggested

I’m surprised that there doesn’t seem to be missing data in patient level characteristics – other than exclusions due to lack of inpatient LOS and stroke classification mentioned in Figure 1. It would be worth clarifying as this seems “too good to be true” for audit data.

Yes there is minimal missing data in the datasets we used in this paper thanks to the patient-level SSNAP data being entered by clinical teams onto a secure webtool with real time data validations to ensure data quality. We mention this in the first line of the first paragraph of the *Data sources and participants* section on page 5. Moreover, we discuss the impact of the minimal missing data bottom of page 7 and top of page 8.

“Regarding the impact of missing data, we assessed the mean change in the outcome variable between the ESD and non-ESD groups in relation to missing predictor data (which amounted to a loss of 0.6% of the total sample size); no substantial differences were found as we observed a mean difference of less than 0.3%.”

Discussion/conclusions:

Overall well written and salient.

Inclusion of an aim and subsequent conclusion regarding “to address inequality and inform evidence based improvements in stroke care” seems an over-call for this data and the analysis performed. There is little specific data, analysis or interpretation regarding inequality and I would suggest removing this from the aims and conclusions.

Thank you – we have removed the two references to inequality from the paper as follows:

Last paragraph of the introduction:

“The study reported here, and in a parallel paper exploring the effectiveness of ESD service models, investigates if trial-based benefits of ESD are realised in practice, with the aim to inform evidence based improvements”

First paragraph of the discussion:

“This study has investigated the effect of ESD on length of hospital stay in real world settings, with the aim to inform evidence based improvements in stroke care.”

On the other hand, the conclusion could be a little more blunt – this study provides concerning data that in contrast to one of the major aims and purported benefits of ESD, the length of stay in the real world appears to be longer! You seem to have under-emphasised this. This reinforces the importance of investigating whether the improved outcomes and client experience also differ between the trials and the real world, and perhaps re-investigating the cost implications.

Thank you - we agree with this comment especially the fact it reinforces the importance of investigating whether outcomes in the real world relate to trial evidence. We've adapted our concluding paragraph with the following amendment:

“ESD was not associated with a reduction in length of hospital stay as previously reported in clinical trials, although the increased average length of stay was just 1 day. This highlights the importance of investigating whether trial-based outcomes of interventions are realised in the real world.”

We have been careful not to over emphasise this finding because the trial-based benefits of ESD were not only a reduction in length of hospital stay but also reduced risk of dependency. As we weren't able to evaluate the benefits of accessing ESD over the longer term we need further studies to fully address this issue.

Reviewer: 2

Comments to the Author

It would be nice to find the absolute number of inpatient days of the ESD and non-ESD group in the text and not only in the tables.

Thank you - we provide this detail in the text on page 10 (second paragraph in the results section):

“Before adjustment, ESD patients had a longer length of hospital stay than non-ESD patients; median 6.9 (interquartile range: 2.9-18.8) compared to 6.0 days (interquartile range: 2.2-24.8)”

and on page 11 (third paragraph in the results section):

“Before adjustment, ESD patients were recorded as having a longer length of stay in hospital than non-ESD patients; median 7.7 (interquartile range: 3.2-23.8) compared to 6.3 days (interquartile range: 2.5-22.1) in 2013-2014, and 6.8 (interquartile range: 2.9-18.5) compared to 5.7 days (interquartile range: 2.1-24.6) in 2015-2016”

Reviewer: 3

Comments to the Author

This is a very well written paper aiming to investigate the impact of Early Supported Discharge (ESD) on length of hospital stay in real world conditions.

I recommend it for publication without any revisions.

Thank you